# Heparin and non-anticoagulant heparin attenuate histone-induced inflammatory responses in whole blood

**John Hogwood**[1,2]*, **Simon Pitchford**[2], **Barbara Mulloy**[2], **Clive Page**[2], **Elaine Gray**[1,2]

**1** National Institute for Biological Standards and Control, South Mimms, Ridge, Herts, United Kingdom,
**2** Sacker Institute of Pulmonary Pharmacology, Institute of Pharmaceutical Science, King's College London, London, United Kingdom

* john.hogwood@nibsc.org

**Data Availability Statement:** All relevant data are within the paper and raw data for the figures are in the supporting information.

## Abstract

Cytotoxic and pro-inflammatory histones are present in neutrophil extracellular traps (NETs) and are elevated in blood in several inflammatory conditions, sepsis being a major example. Compounds which can attenuate activities of histones are therefore of interest, with heparin being one such material that has previously been shown to bind to histones. Heparin, a successful anticoagulant for nearly a century, has been shown experimentally to bind to histones and exhibit a protective effect in inflammatory conditions. In the present study carried out in whole blood, heparin and selectively desulfated heparin reduced histone induced inflammatory markers such as interleukin 6 (IL 6), interleukin 8 (IL 8) and tissue factor and C3a, a complement component. The selectively desulfated heparins, with reduced anticoagulant activities, retained a high degree of effectiveness as an anti-histone agent, whereas fully desulfated heparin was found to be ineffective. The results from this study indicate that the presence of sulfate and other specific structural features are required for heparin to attenuate the inflammatory action of histones in whole blood.

## Introduction

There is interest in the role that heparin can play beyond acting as an anticoagulant and antithrombotic. A range of *in vitro* studies have demonstrated that heparin possesses beneficial anti-inflammatory and anti-complement activity [1]. It is conceivable that the utilisation of heparin in many indications may not require its anticoagulant activity and this anticoagulant activity could be considered an undesirable feature in such settings [2]. There are now a number of investigations [3–6] that focus on non-anticoagulant heparins which have been produced by chemical modifications. These modified heparins have been used to identify some of the structural requirements for indications unrelated to the well described anticoagulant and antithrombotic actions of heparin. However, there is a clear need for simple, robust assays that reflect the non-anticoagulant actions of heparin to aid the development of these modified heparins as potential therapeutics.

**Funding:** The authors received no specific funding for this work.

**Competing interests:** The authors have declared that no competing interests exist.

One area receiving some attention is the ability of heparin to act as an anti-inflammatory agent [7] with the proinflammatory activity of neutrophils investigated as one potential target for heparin [8, 9]. A rationale for this is the essential role that neutrophils play in host immunity providing protection against microbial infection, however, inappropriate or exaggerated activation of neutrophils can contribute to diseases such as psoriasis, chronic obstructive pulmonary disease (COPD) and in the complications of sepsis [10]. Neutrophils also release neutrophil related extracellular traps (NETs), as an immune response, via a form of cell death termed NETosis [11, 12]. Of the components in NETs, histones are in high abundance contributing to over 70% of the total protein mass [13] and histones are known to have both antimicrobial and cytotoxic properties [14].

In addition to their presence in NETs, extracellular histones have been shown to be markers of disease, released from cells as damage-associated molecular pattern proteins [15, 16]. Their presence has been correlated with thrombocytopenia [17], organ failure [18] and severity of sepsis [19]. Histones have also been demonstrated to induce *in vitro* tissue factor expression in both endothelial cells [20] and monocytes [21]. Agents which could be used to reduce histone levels would therefore potentially be beneficial in the clinic as a potential treatment of sepsis and therefore it is of interest that heparin has been shown to bind to histones [22].

Heparin is commonly used in the treatment of sepsis where its role is considered to be beyond anticoagulation [23]. Part of this role may be due to its known ability to interact with neutrophils and disrupt their activity, attenuating elastase release and neutrophil aggregation [8]. Heparin has also been shown to reduce the inflammatory effects induced by histone administration [17, 19, 24]. The anticoagulant properties of heparin are undesirable in this setting. It has been demonstrated that heparin with its anticoagulant component removed by fractionation retained its cytoprotective effects *in vitro* following histone addition [14]. In an *in vivo* acute injury survival model characterised by elevated histone levels, N-acetyl-heparin, an apparent non-anticoagulant heparin was observed to be almost as effective as unmodified heparin in improving survival rate [25]. These investigations indicate heparin can be developed as an anti-inflammatory agent without its anticoagulant activity.

The main aims of the current work were to study the structural requirement, especially the locations of sulfation, for heparin to elicit its anti-inflammatory activities and to investigate whether stimulation of whole blood with histones, could be used as an assay to quantify the anti-inflammatory effects of heparins. Heparin is a highly negatively charged polysaccharide, heavily substituted with sulfate groups. Some reports have indicated that the anti-inflammatory properties of heparin may be related to its sulfate content [26, 27]. The main repeating trisulfated disaccharide unit of heparin consists of 2-O-sulfated iduronic acid alternating with N-sulfated, 6-O-sulfated glucosamine. In this study, modified heparins with selected sulfate groups (2-O-sulfate, 6-O-sulfate or N-sulfate) removed and, in the case of N-sulfate removal, re-N-acetylated were investigated to provide an insight into the importance of the location of the sulfate groups. A whole blood model was considered preferable as it includes the range of blood components previously shown to be affected by histones including monocytes [21], neutrophils [14], platelets [28] and red blood cells [29]. This is an important consideration for any assay as it has been shown that inflammatory responses in whole blood differ from the response of individual blood components when studying the effect of lipopolysaccharides [30, 31]. Furthermore, there are no published data on the anti-histone effects of heparin and related compounds in a whole blood system. In the present study concurrent measurement of several markers involved in inflammation were analysed: tissue factor, IL6 and IL8. As heparin also interacts with the complement pathway [32], complement C3a was also measured to investigate the link between the complement and inflammatory systems. The modified heparins with precisely estimated anticoagulant activity (information lacking in other studies), expressed in

International Units, were compared with unmodified heparin to determine potential structural requirements for heparin to exhibit anti-histone / anti-inflammatory activity in whole blood.

## Methods

### Heparins

Two selectively desulfated heparins, 2-O-desulfated and N-desulfated-re-N-acetylated, were prepared from a single batch of heparin. The 2-O-desulfated heparin was prepared as described by Jaseja [33] and N-Acetyl heparin was prepared as described by Nagasawa [34]. Other modified heparins were 6-O-desulfated heparin (Iduron, Alderley Edge, UK), N-desulfated heparin and N-acetyl-de-O-sulfated heparin (Sigma, Gillingham UK). Confirmation of sulfate modification of all samples was carried out by NMR analysis with chemical shifts compared to published data [35, 36]. A clinical heparin sample (Wockhardt Ltd, Wrexham, UK) was also included in some assays.

### Anticoagulant activity of modified heparins

The anticoagulant activities of the modified heparins were estimated by several different methods: antithrombin dependent anti-IIa (European and United States Pharmacopeia method for potency assignment of clinical heparin products), antithrombin dependent anti-Xa, heparin cofactor II dependent anti-IIa and plasma clotting (activated partial thromboplastin time using human plasma) assays were carried out as previously described [37]. All estimations of activity were calculated against the 6[th] International Standard for Unfractionated Heparin (07/328, NIBSC, UK) using parallel line model in CombiStats 5.0 (EDQM, Strasbourg, France).

### Whole blood with histones

Whole blood (10 ml) was collected from healthy volunteers (ethical approval obtained from National Institute for Biological Standards and Control's Human Materials Advisory Committee) by venepuncture from an antecubital vein into a sterile syringe. Blood was anticoagulated with recombinant hirudin (Pharmion, UK) to a final concentration of 30 μg/ml. Blood was processed for experiments within 30 minutes of collection.

Whole blood was distributed into microtitre plates (Greiner Bio-one, Frickenhausen, Germany) to which increasing concentrations of histones (Type III-S, Sigma, Gillingham, UK) were added. The microtitre plates were placed on an orbital shaker (500 rpm) in a humidified $CO_2$ incubator and incubated for either 6 or 24 hours. After incubation, the blood was spun, and the supernatant was collected for cytokine analysis. The cellular component was washed four times using Hank's balanced salt solution before being frozen for tissue factor analysis.

### Effect of histones and heparin on whole blood

A fixed concentration of histones (50 μg/ml final level) was added to whole blood, and after 5 minutes different amounts in International Unit (IU) of a clinical unfractionated heparin were added. Modified heparins were used in μg/ml (final) concentrations as specified in the results. Whole blood was incubated as above for 6 hours prior to collection of supernatant and cellular components for analysis. Statistical analysis of the effect of each heparin was carried out by using analysis of variance (ANOVA) to fit the data to a general linear model. Dunnett's multiple comparison, which corrects for donor to donor variability through use of a control sample (histone only response for each donor) was then used to compare results.

### Analysis of inflammatory markers

Measurement of plasma levels of IL6, IL8 and complement factor C3a were carried out by ELISA following the manufacturer's instructions (Thermo Fisher, Hemel Hempstead UK). The level of tissue factor was measured by a clotting assay following three freeze-thawing cycles of the cells [38]. The clotting assay was carried out by mixing each freeze-thawed sample with an equal volume of pooled platelet-poor normal plasma (National Blood and Transfusion Service, Colindale, UK) and incubated at 37°C in Ceveron Ten coagulometers (Technoclone, Vienna, Austria) before addition of an equal volume of 25 mM calcium chloride. The obtained clotting times were converted into tissue factor units against a tissue factor (14/238, NIBSC, UK) standard curve with values reported in arbitrary units. As described above statistical analysis of data was carried out using Dunnett's multiple comparison with the histone only response as the control sample.

## Results

### Effect of histones on whole blood

The ability of histones to induce an inflammatory response in whole blood from six different donors was assessed after 6- and 24-hour incubation (Fig 1). After a 6-hour incubation period, histones induced production of the inflammatory markers IL6, IL8 and tissue factor (TF), and increased complement activation as observed by increased C3a levels over background responses. After 24-hours, the levels of both IL6 and IL8 had reached the same level across all concentrations of histones and were higher than the samples that had been incubated for 6-hours. There were concentration dependent increases in both C3a and TF levels following 6- and 24-hour incubation periods, with levels after 24 hours generally being higher for C3a but lower for TF.

### Ability of heparin to influence histone responses in whole blood

The addition of unmodified heparin to whole blood, in the absence of histones, had a mild attenuating effect on the background levels of C3a, IL8 and TF (see Fig 2). C3a was lowered

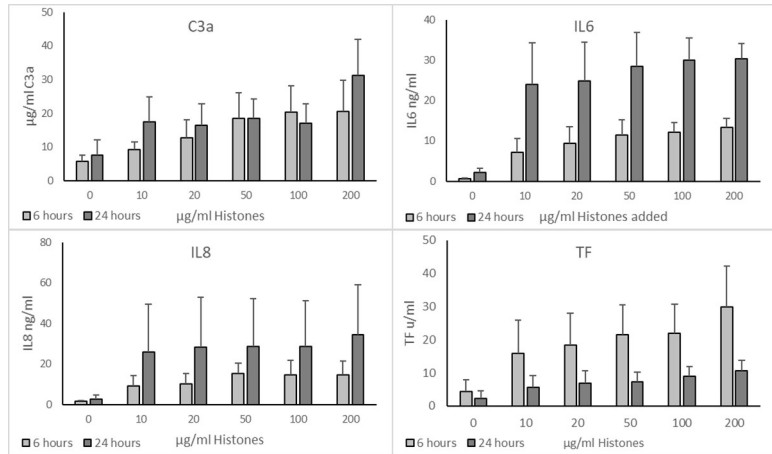

**Fig 1. Effect of histones on inducing inflammatory markers in human whole blood over 6 and 24 hours.** Whole blood was stimulated with increasing concentrations of histones over 6 and 24 hours. Released C3a, IL-6 and IL-8 were measured in plasma and tissue factor was measured from the cellular component. For each timepoint all concentrations were $p < 0.001$ relative to the background (0 μg/ml) reading by Dunnett's multiple comparison. Error bars = standard deviation of the average from 6 donors.

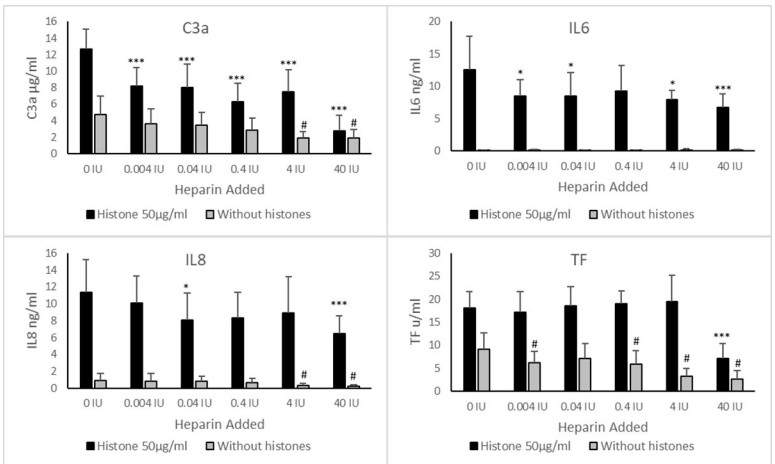

**Fig 2. Effect of heparin on background and histone induced inflammatory responses in whole blood.** Increasing concentrations of heparin (in International Units) were added to whole blood treated with/without 50 μg/ml histones and incubated for 6 hours. Released C3a, IL-6 and IL-8 were measured in plasma and tissue factor was measured from the cellular component. Analysis of each group (with/without histones) by Dunnett's multiple comparison was carried out using the background (0 IU) as the control group, * or # = p < 0.05, ** or ## = p < 0.01, *** or ### = p <0.001. Error bars = standard deviation of the average from 6 donors.

from 4.8 ±2.2 μg/ml to 1.9 ±1.0 μg/ml, IL8 from 0.9 ±0.8 ng/ml to 0.2 ±0.1 ng/ml and TF from 9.2 ±3.5 u/ml to 2.7 ±1.8 u/ml following incubation with 40 IU/ml heparin. There was no effect on the level of IL6 measured.

Histones at 50 μg/ml were used to determine the effect of heparin on attenuating the histone-induced inflammatory response in whole blood (Fig 2). This concentration is within the range observed in patients with sepsis [39]. A clinical heparin sample added at different concentrations in IU, was able to reduce the effect of histones on whole blood. The reduction in C3a levels was significant ($p < 0.001$) for all concentrations, with the histone-induced response reduced from 12.7 ±2.4 μg/ml to 8.2 ±2.3 μg/ml by 0.004 IU/ml heparin, and to below the background level at 40 IU/ml heparin, 2.8 ±1.9 μg/ml.

All concentrations of heparin reduced the level of IL6 generated; histones alone gave 12.5 ±5.2 ng/ml which was reduced to between 6.7 ±2.1 and 9.2 ±4.0 ng/ml by heparin. At the midpoint heparin concentration, 0.4 IU/ml, there was no difference to the histones only response ($p = 0.091$), whilst the other concentrations above and below this were significantly different (0.004 IU/ml was $p = 0.030$, 0.04 IU/ml was $p = 0.027$, 4 IU/ml was $p = 0.012$, 40 IU/ml was $p = 0.001$). The histone induced IL8 response was also reduced by heparin, from 11.4 ±3.8 ng/ml for histone only to 10.1 ±3.2 ng/ml with 0.004 IU/ml heparin and further reduced to 6.4 ±2.2 ng/ml when 40 IU/ml heparin was added. These reductions were only significant at the concentrations of 40 IU/ml ($p = 0.001$) and 0.04 IU/ml ($p = 0.035$) which was in part due to high variability of donor responses and were reflected in the wide error bars. Only 40 IU/ml heparin was able to affect the level of TF generated, which was reduced from 18.1 ±3.6 u/ml to below background at 7.2 ±3.1 u/ml ($p < 0.001$).

### Effect of selectively desulfated heparin

The unfractionated heparin sample used to produce the 2-O-desulfated and N-acetyl-heparin had a specific activity of 209 IU/mg by the potency assignment (anti-IIa) method (Table 1). The anti-Xa activity for this heparin was broadly the same, 214 IU/mg, giving an aIIa/aXa ratio of 0.98. The clotting assay gave an activity of 209 IU/mg, and the HCII activity was 241 IU/mg.

**Table 1. Anticoagulant activities of modified heparin samples, estimated against.** The 6[th] IS Unfractionated Heparin, 07/328, in IU/mg with 95% confidence limits in brackets.

| | IU/mg (95% Confidence Limits) | | | |
|---|---|---|---|---|
| Heparin | Plasma Clotting | Anti-Xa | Anti-IIa | HCII |
| Parent | 209 (191–229) | 214 (207–221) | 209 (196–223) | 241 (223–261) |
| 2-de-O-sulfated | 80.5 (76.2–85.0) | 71.2 (68.0–74.6) | 43.4 (40.5–46.5) | 14.5 (13.2–15.9) |
| N-acetyl | 5.1* (4.5–5.7) | <1 | <1 | 1.1 (1.0–1.2) |
| 6-de-O-sulfated | 18.2 (17.5–18.9) | 28.4 (26.4–30.6) | 14.9 (14.2–15.7) | 17.1 (15.7–18.7) |
| De N-sulfated | <1 | <1 | <1 | <1 |
| N-acetyl-de-O-sulfated | <1 | <1 | <1 | <1 |

Plasma clotting = APTT clotting assay using human plasma, Anti-Xa = antithrombin dependent anti-factor Xa assay, Anti-IIa = antithrombin dependent anti-factor IIa assay, HCII = heparin cofactor II dependent anti-factor IIa assay. Values have been calculated using multiple dilution models–slope ratio for the clotting assay and parallel line for other assays. Assays results were considered valid except for those indicated and where activity was below quantifiable limits.

* Sample response was non-linear to standard across concentration response range

Removal of the 2-O-sulfate from heparin reduced anticoagulant activity in all assays with the greatest reduction observed in the HCII dependent assay to 14.5 IU/mg, and the least reduction was by the plasma clotting assay, 80.5 IU/mg. Removal of the N-sulfate from heparin produced a material which had no measurable activity in any assay. Heparin which had been de-N-sulfated and then re-N-acetylated (N-acetyl) had very low activity with no measurable antithrombin mediated anti-Xa and anti-IIa activity, some minor HCII activity at 1.1 IU/mg and limited plasma-based clotting activity (5.1 IU/mg), though the latter response was not parallel to the standard indicating a different mechanism of action to the heparin standard. Compared to unmodified heparin, the 6-O-desulfated heparin had reduced anticoagulant activity assessed by plasma clotting, whilst antithrombin anti-IIa and HCII dependent assays were broadly similar, 18.2 IU/mg, 14.9 IU/mg and 17.1 IU/mg with a slightly higher value by anti-Xa, 28.4 IU/mg. A fully desulfated heparin, with re-N-acetylation had no measurable anticoagulant activity.

A concentration of 200 µg/ml of the modified heparin was used in the histone-whole blood assay, based on an arbitrary assumption that 40 IU/ml clinical heparin equates to approximately 200 µg/ml (200 IU/mg specific activity). This high level of heparin is similar to that used by other groups in *in vitro* experiments to attenuate the effect of histones [17, 19]. At 200 µg/ml unmodified heparin (Fig 3) was able to reduce the level of all the markers measured–C3a, IL6. IL8 and TF to a significant degree (all $p < 0.001$), with both C3a and TF attenuated to below the background level (3.6 ±2.7 µg/ml and 5.0 ±4.7 u/ml respectively) and IL6 by 45% (from 13.5 ±4.5 to 7.4 ±2.4 ng/ml) and IL8 by 70% (17.5 ±5.9 to 6.7 ±3.6 ng/ml).

Modification of heparin by N-desulfation or full desulfation with N-acetylation removed the ability of heparin to attenuate the histone effects on whole blood by the measured analytes. The other desulfated heparins (2-O-desulfated, N-acetyl and 6-O-desulfated) showed an ability to reduce the effect of histones, but this was lower than unmodified heparin. Measurement of C3a and TF showed that the 2-O-desulfated heparin could attenuate histone-induced responses more effectively than N-acetyl heparin which itself was more effective than the 6-O-desulfated heparin (C3a was 7.8 ±3.9, 10.1 ±5.3, 9.8 ±4.9 µg/ml and TF was 5.9 ±5.0, 12.4 ±9.5 and 14.5 ±10.5 u/ml–respectively 2-O-desulfated, N-acetyl and 6-O-desulfated heparin). The reduction in the level of IL6 was broadly the same for the three samples (from histone only 13.5 ±4.5 to 9.1 ±2.0, 8.2 ±2.2 and 10.6 ±4.6 ng/ml respectively). For IL8 the 2-O-desulfated

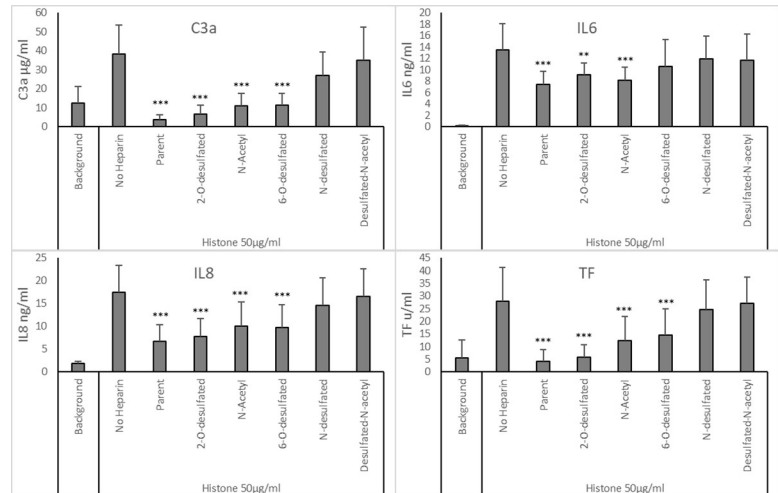

**Fig 3. Ability of selectively desulfated heparin to attenuate histone induced inflammatory responses in whole blood.** Histones, 50μg/ml, and differently desulfated heparins, 200 μg/ml, were added to whole blood and incubated for 6 hours. Released C3a, IL-6 and IL-8 were measured in plasma and tissue factor was measured from the cellular component. Analysis of data was carried out using Dunnett's multiple comparison with no heparin (histones only) as the control sample, * = $p < 0.05$, ** = $p < 0.01$, *** $p < 0.001$. Error bars = standard deviation of the average from 6 donors.

heparin reduced the response to 7.8 ±3.9 ng/ml from 17.5 ±5.9 ng/ml, whilst N-acetyl and 6-O-desulfated heparin reduced histone-induced responses to similar levels of 10.1 ± 5.3 ng/ml and 9.8 ±4.9 ng/ml respectively. All these reductions were statistically significant ($p < 0.01$) compared to the absence of heparin.

## Effect of N-acetyl heparin on histone-induced responses

With an anticoagulant activity below 10 IU/mg, N-acetyl heparin was investigated further at lower concentrations. The parent heparin used to prepare the sample was included for comparison. Five concentrations were used, 0.02, 0.2, 2, 20 and 200 μg/ml (Fig 4). Only the highest concentration of N-acetyl heparin was able to significantly reduce the effect of histones on whole blood for all inflammatory mediators measured–C3a from 22.0 ±8.3 μg/ml to below background at 5.5 ±2.0 μg/ml, IL6 from 8.8 ±3.3 to 5.8 ±3.2 ng/ml, IL8 from 12.9 ±6.6 ng/ml to 6.2 ±0.8 ng/ml and TF from 39.9 ±7.5 to 16.1 ±7.0 u/ml ($p < 0.01$). Attenuation by unmodified heparin at 200 μg/ml was greater than that by N-acetyl heparin for all inflammatory mediators—C3a was– 2.0 ±0.9 μg/ml, IL6 was 4.7 ±1.4 ng/ml, IL8 was 4.2 ±0.9 ng/ml and TF was 5.7±4.8 u/ml ($p < 0.01$).

Lower concentrations of both N-acetyl and parent heparins were less effective at attenuation of the histone-induced response. For C3a, a significant reduction was observed when 20 ($p = 0.001$) and 2 μg/ml ($p = 0.022$) were used, with no difference to background at 0.2 and 0.02 μg/ml ($p = 0.226$ and $p = 0.472$ respectively). For IL6, with the exception of 0.02 μg/ml, all higher concentrations generated a small but significant ($p < 0.01$) reduction of IL6 –for N-acetyl heparin values were 7.0 ±2.9, 6.8 ±2.7 and 6.1 ±2.6 ng/ml respectively and for the parent heparin, levels were lowered to 6.5 ± 3.2, 5.8 ±2.3 and 4.9 ±2.1 ng/ml respectively compared to 8.8 ±3.3 ng/ml for histone only. The effect of both heparins on IL8 levels was similar to IL6 (Fig 4). For TF, lower concentrations (<200 μg/ml) of N-acetyl heparin did not reduce the effect of histones on whole blood, whilst for the parent heparin both 20 and 2 μg/ml were able to lower TF levels to 32.5 ± 5.7 and 35.7 ± 8.7 from 39.7 ± 7.5 u/ml ($p < 0.05$).

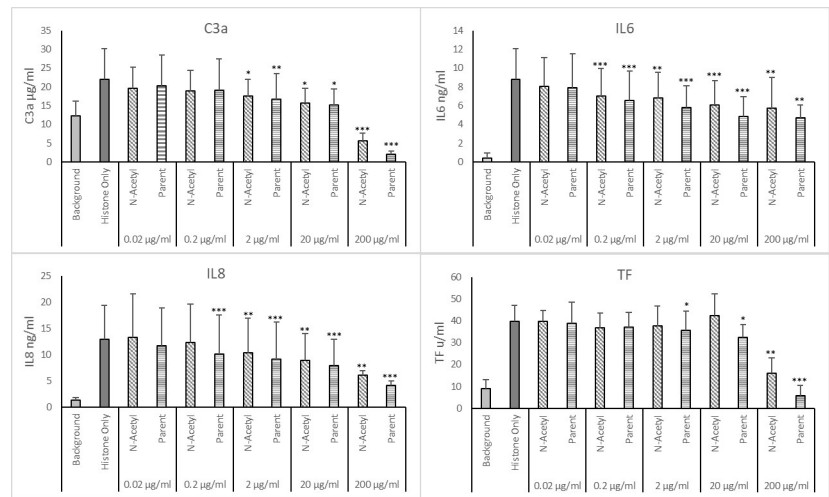

**Fig 4. Dose effect of heparin and N-acetyl heparin on attenuating histone induced inflammatory responses in whole blood.** Histones, 50µg/ml, and increasing concentrations of parent heparin and n-acetyl-heparin were added to whole blood and incubated for 6 hours. Released C3a, IL-6 and IL-8 were measured in plasma and tissue factor was measured from the cellular component. Analysis of data was carried out using Dunnett's multiple comparison with no heparin (histones only) as the control sample, * = p < 0.05, ** = p < 0.01, *** p < 0.001. Error bars = standard deviation of the average from 6 donors.

## Discussion

The ability of heparin to interact with a wide range of proteins has been well established [40]. Despite these known interactions, development and approval of heparin in a therapeutic role beyond anticoagulation has been slow. Many studies indicate that heparin interacts with proteins implicated in inflammatory responses and can influence their roles in both *in vivo* and *in vitro* settings, primarily acting in a beneficial anti-inflammatory manner [2]. Indeed, the role of heparin in the treatment of sepsis, a condition that results in elevated levels of histones [39], is considered to be beyond that of an anticoagulant [23]. These broad anti-inflammatory effects may explain why heparin has been successfully used to treat some inflammatory conditions [2]. The results presented here demonstrated the ability of heparin and modified heparins to act as an anti-inflammatory agent to reduce the cytotoxic effects of histones when added to whole blood.

Previously, extracellular histones were shown to increase TF secretion/expression in peripheral monocytes [21] and endothelial cells [20], and this was confirmed when using whole blood in this study. Further biomarkers measured in whole blood, IL6, IL8 and complement C3a were all increased by histones in a concentration dependent manner. These responses support the published observation of increased TNFα and IL6 expression in endothelial cells [19] following incubation with histones. The activation of the complement system by histones, as measured in this work by C3a, could explain the increase in survival of C5 deficient mice over wild type mice following histone administration [41]. Landsem et al have reported on this complement-inflammation axis, linking complement activation with elevation of TF expression in *Escherichia coli* stimulated human whole blood [42].

In the present study, a clinical heparin preparation was able to lower the levels of the inflammatory markers measured in histone treated whole blood, supporting the observations that heparin reduces histone-induced cell death of endothelial cells and leukocytes *in vitro*, and improves survival in an *in vivo* model [19, 20, 24]. This is the first study that has used a whole blood method to demonstrate the attenuating effect of heparin on the inflammatory

response generated by histones. The effect of heparin as an anti-histone agent was concentration dependent with a clear effect on C3a levels, and variable effects on IL6 and IL8 levels primarily due to differences in donor to donor response which influenced statistical validity at some heparin concentrations. Only the highest level of heparin (200 μg/ml) used in this study was able to reduce the level of TF induced by histones in whole blood, which contrasts to the observation that TF levels from monocytes were reduced when incubated with plasma from patients given therapeutic levels of heparin versus untreated subjects [21].

As described in the introduction, the bleeding side effect of heparin can be a drawback for heparin to act as an anti-inflammatory agent as a much higher dose would be needed for it to be effective for this indication. Thus, modification of heparin to reduce or remove anticoagulant activity is an attractive approach to improve its therapeutic potential as an anti-histone / anti-inflammatory agent. To investigate this, heparins modified by selective desulfation, with reduced or no measurable anticoagulant activity were studied. Of the selectively desulfated heparin preparations investigated, two retained some anticoagulant activity (the 2-O-desulfated and 6-O-desulfated heparin), whilst others had no measurable anticoagulant activity (N-desulfated, N-acetyl and fully desulfated heparin). Although the concentration used for comparison was high, at 200 μg/ml, the level was chosen to allow for comparison to the clinical heparin used in this study as this level demonstrated a statistically significant reduction on histone-induced responses in all assays used. The removal of sulfates at different sites in heparin reduced its relative effectiveness (compared to unmodified heparin) as an anti-histone treatment, whilst complete sulfate removal rendered heparin unable to act as an anti-histone agent, demonstrating the importance of negatively charged sulfate groups. One subtle structural feature was revealed by the re-N-acetylation of the N-desulfated heparin. The elimination of the N-sulfate removed the anti-histone activity, whilst the acetylation of this site (producing N-acetyl heparin) restored anti-histone function. This was probably due to the exposure of the amine group in heparin following N-desulfation. The positively charged amine group may disturb the charge-based interaction of heparin with histone which is then restored by the presence of the neutral acetyl group. Of the sulfate sites studied, removal of the 2-O-sulfate retained some 90% of the anti-histone activity of unmodified heparin. However, the anticoagulant activity for 2-O-desulfated heparin is still high at ~80 IU/mg, thus it may not be a viable material to act purely as an anti-histone agent. The 6-O-desulfated heparin at ~18 IU/mg reduced the C3a level to 75% of that observe for the unmodified heparin.

The responses observed with the modified heparins on levels of IL6, IL8 and tissue factor showed a similar importance for the sulfate sites studied, though the absolute reductions relative to histone only were lower when compared to the attenuation of C3a levels by the different heparins. It has been shown that the sulfation level of heparin is crucial for its ability to influence the complement system, with over-sulfation enhancing this ability [43], and sulfate loss reducing it [44]. Therefore, it is reasonable to assume that the sulfation level of heparin influences the degree by which heparin inhibits histone-induced C3a or by direct interaction with complement factors [44, 45].

Of the materials prepared, the N-acetyl heparin has the greatest potential as a therapeutic anti-histone agent due to minimal anticoagulant activity. When compared with parent heparin at the same concentration, the N-acetyl heparin only had a slightly reduced anti-histone activity. The relative effectiveness of N-acetyl heparin on dampening histone-induced responses was comparable to heparin when IL6 and C3a were measured but showed reduced effectiveness for IL8 and tissue factor.

The mechanism by which histone damage can be prevented by heparin can in part be explained by the likely interaction between heparin and histones. Histones have been shown to bind to heparin [46, 47] and will also bind to heparan sulfate, a less sulfated polysaccharide

with structural similarities to heparin [40]. It is the binding of histones to heparan sulfate which forms part of the mechanism by which extracellular chromatin (and histones) are cleared [48] from circulation. In an acute lung model, histones accumulated on heparan sulfate within the lung giving rise to lung injury [49] with treatment using heparinase or infusion with heparin providing a protective benefit. It would be reasonable to assume, that heparin possessing a high negative charge (via its many sulfates) interacts with these positively charged regions on the surface of the histone molecule [50, 51]. Changes in heparin sulfation pattern would therefore alter the interaction between histones and heparin, with sulfate loss reducing interaction as demonstrated in this study. The same assumption might explain why heparin competes successfully with the less densely sulfated cell surface heparan sulfates for binding to histones [49].

A non-anticoagulant heparin has the potential to be a useful therapeutic agent for the treatment of diseases, such as acute lung injury and sepsis, which results in elevated histone levels in blood [25, 39]. This study and previous work [14, 24] demonstrate that heparin attenuates the harmful effects of histones on cells. This work has shown that modification of heparin to remove anticoagulant activity (measured by the potency assignment method for clinical heparins, the anti-IIa assay) maintains the potential benefit of heparin, albeit with a slight reduction in effectiveness. Whilst effective concentrations of N-acetyl heparin were high (>20 µg/ml), this is in the context of an anticoagulant to which this concentration is being compared and may not be considered high for a non-anticoagulant heparin to act as an anti-histone or anti-inflammatory agent.

## Summary

Using a whole blood model, we have demonstrated the pro-inflammatory and complement activation effects of histones. This approach differs from previous published work using isolated cells such as monocytes in that a whole blood system was used, allowing the interaction of histones and heparins with all blood components thereby providing a better simulation of the *in vivo* setting. Our work has brought together the measurement of several relevant inflammatory markers (IL6, IL8, tissue factor and complement C3a) previously only studied in individual cell types or in animal models. The results showed that unmodified heparin and modified heparins were able to attenuate inflammation, and our results suggest that a whole blood assay may provide a good platform to study the complex anti-inflammatory effects of heparin and related products.

The use of modified and selectively desulfated heparin has provided new insight into the structure and function relationship between heparin and histones. The reduction of anti-histone activity by the removal of the N-sulfate group highlighted the importance of heparin's negative charge. N-acetylation restored the activity to the N-desulfated heparin suggesting that the "shielding" of the positively charged amine group exposed by N-desulfation is important for the activity. It is clear that the presence of sulfates was also important as the completely desulfated, but N-acetylated heparin did not show any anti-histone activity. However, the position of the sulfate may not be critical as the removal of either the 2-O, the 6-O or the N-sulfate groups (when replaced by the neutral acetyl group) all had minimal effect on the inflammatory markers measured. The disruption of anti-histone activity by interference with the charge distribution of heparin is consistent with an electrostatic interaction between the negatively charged polysaccharide and the positively charged histones.

## Supporting information

**S1 Raw data. Raw data used to create Figs 1 to 4.**
(XLSX)

## Acknowledgments

The authors would like to thank Peter Rigsby (National Institute for Biological Standards and Control, UK) for statistical advice.

## Author Contributions

**Conceptualization:** John Hogwood, Barbara Mulloy, Clive Page, Elaine Gray.

**Data curation:** John Hogwood.

**Formal analysis:** John Hogwood, Elaine Gray.

**Investigation:** John Hogwood.

**Project administration:** John Hogwood, Clive Page, Elaine Gray.

**Supervision:** Simon Pitchford, Barbara Mulloy, Clive Page, Elaine Gray.

**Writing – original draft:** John Hogwood.

**Writing – review & editing:** John Hogwood, Simon Pitchford, Barbara Mulloy, Clive Page, Elaine Gray.

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
