## [Decision Letter · Decision Letter 0]

25 Feb 2020

PONE-D-20-00191

Heparin and non-anticoagulant heparin attenuate histone-induced inflammatory responses in whole blood

PLOS ONE

Dear Mr Hogwood,

Thank you for submitting your manuscript to PLOS ONE. After careful consideration, we feel that it has merit but does not fully meet PLOS ONE’s publication criteria as it currently stands. Therefore, we invite you to submit a revised version of the manuscript that addresses the points raised during the review process.

The authors should respond to the specific concerns of both reviewers, including the number of samples used in the study and the possible effects of anticoagulants on the observed determinations. The authors should also include a more precise presentations of the novel aspects of the present study as compared to those in the literature.

We would appreciate receiving your revised manuscript by Apr 10 2020 11:59PM. To enhance the reproducibility of your results, we recommend that if applicable you deposit your laboratory protocols in protocols.io, where a protocol can be assigned its own identifier (DOI) such that it can be cited independently in the future. For instructions see: http://journals.plos.org/plosone/s/submission-guidelines#loc-laboratory-protocols

We look forward to receiving your revised manuscript.

Kind regards,

Pablo Garcia de Frutos

Academic Editor

PLOS ONE

Journal Requirements:

Reviewers' comments:

Reviewer's Responses to Questions

**Comments to the Author**

1. Is the manuscript technically sound, and do the data support the conclusions?

Reviewer #1: Partly

Reviewer #2: Yes

2. Has the statistical analysis been performed appropriately and rigorously? 

Reviewer #1: I Don't Know

Reviewer #2: Yes

3. Have the authors made all data underlying the findings in their manuscript fully available?

Reviewer #1: Yes

Reviewer #2: Yes

4. Is the manuscript presented in an intelligible fashion and written in standard English?

Reviewer #1: Yes

Reviewer #2: Yes

5. Review Comments to the Author

Reviewer #1: Heparin and non-anticoagulant heparin attenuate histone-induced inflammatory responses in whole blood

General comments

The manuscript by Dr Hogwood et al describes data from an in vitro assay that employs hirudin-anticoagulated whole blood to measure in vitro histone-mediated effects on inflammation and coagulation. Further the manuscripts describes how use of heparin and a number of desulfated glycans influences the measurements, in particular coagulation and histone-mediated effects on inflammation are described.

The manuscript is interesting and readable but there are a number of issues, as explained below, that need further attention.

The increment in knowledge by this study is marginal, as it confirms several points in literature (such as the effects of heparins) and the effects of histones on inflammation. What the added value of these data are, is not made clear. In vivo data addressing this topic have been published already. The fact that the dampening effect of heparin on the cell toxicity of histones does not depend on the presence of anticoagulant properties of histones has already been described and use of a non-anticoagulant heparin to reduce histone-mediated toxicity is not novel.

Specific Points

• A lysine-rich fraction of histones is used, while it is known that, arginine-rich histones H3 and H4 have a higher inflammatory and cytotoxic potency.

• A limited number of different samples were tested (n=6), this sample size seriously limits significance of the study.

• The study of the effects of different glycans is only performed in the context of histones, where the direct effects of for instance heparin remain incompletely addressed. If complexes are formed between histones and heparin, these may cause the effects observed, but likewise the heparin itself may also be responsible for observed effects.

• In addition to this latter point, it has not been studied whether or not these complexes are formed, and only the formation of such a complex is inferred from the data, while it was not shown they actually exist, or that binding occurs and accompanies the effects measured.

• It is unclear why a limited selection of inflammation biomarkers was chosen.

• Anticoagulated blood is used, how did this affect the observations made, as this is non-physiological. No information on NET-formation or cell activation is provided.

Reviewer #2: In this review the authors us an in vitro approach with hirudin collected healthy whole blood to study the impact of heparins on the histone-induced inflammatory response.

A search of the literature did not reveal any similar in vitro experiments. There is some literature in this area in vivo. The current experiments are primarily descriptive.

Concerns

1. Why was the whole blood collected in hirudin? Would the same effects be seen if the blood was collected in citrate?

2. The authors report low levels of cytokines and C3a in response to histone stimulation (ng/ml range). What is the detection range of the ELISA for these markers?

3. The introduction could be shortened and still convey the rationale for the study.

4. Levels of markers should not be reported as negative. Perhaps best to report as percent reduction compared to control.

5. The supplemental figure should be part of the main manuscript.

6. The authors are missing a recent publication that supports their findings Zhu et al J. Trauma Acute Care Surgery 2019

7. Page 2 line 33 is missing a reference, Page 2 line 44 is this a complication of sepsis or pathogenesis (see McDonald et al Blood 2017

6. PLOS authors have the option to publish the peer review history of their article (what does this mean?). If published, this will include your full peer review and any attached files.

Reviewer #1: No

Reviewer #2: No

---

## [Author Response · Author response to Decision Letter 0]

1 Apr 2020

Response to reviewers comments

Reviewer #1: Heparin and non-anticoagulant heparin attenuate histone-induced inflammatory responses in whole blood

General comments

The manuscript by Dr Hogwood et al describes data from an in vitro assay that employs hirudin-anticoagulated whole blood to measure in vitro histone-mediated effects on inflammation and coagulation. Further the manuscripts describe how use of heparin and a number of desulfated glycans influences the measurements, in particular coagulation and histone-mediated effects on inflammation are described.

The manuscript is interesting and readable but there are a number of issues, as explained below, that need further attention.

The increment in knowledge by this study is marginal, as it confirms several points in literature (such as the effects of heparins) and the effects of histones on inflammation. What the added value of these data are, is not made clear. In vivo data addressing this topic have been published already. The fact that the dampening effect of heparin on the cell toxicity of histones does not depend on the presence of anticoagulant properties of histones has already been described and use of a non-anticoagulant heparin to reduce histone-mediated toxicity is not novel.

The novel important finding of this study is the structure and function relationship between heparin and histones. Sulfation was found to be important for activity but the exact positions of the negatively charged sulfates were not critical. We have shown that it is important to maintain the overall negative charge; exposure of the positively charged amine group on heparin disrupted the anti-histone activity of heparin. We apologise that this was not made clear in the original submission and have added new material to the introduction, revised the discussion and expanded our summary to reflect our findings.

Specific Points

• A lysine-rich fraction of histones is used, while it is known that, arginine-rich histones H3 and H4 have a higher inflammatory and cytotoxic potency.

The lysine rich histones were used by other groups to critically demonstrate interaction with heparin which was part of the objective of this work. We agree that it would be important to study H3 and H4 and we intend to carry out these follow up experiments. Thank you for the suggestion.

• A limited number of different samples were tested (n=6), this sample size seriously limits significance of the study.

Unfortunately, we are not clear what this is referring to, the number of heparin samples or the number of donors. If this refers to the different heparin samples (6 used) then the heparins used in this study were chosen for their key structural features. We do intend to extend our study to other modified heparins with other characteristics to further explore structure and function relationship, for example, different molecular weight fractions/fragments. If this refers to the number of donors used, we have been advised by our statistician that the use of multiple dose response and a control sample in conjunction with Dunnett’s multiple comparison and results from previous similar studies that 6 donors would be sufficient to generate statistically valid results that could be used for objective assessment and interpretation. 

• The study of the effects of different glycans is only performed in the context of histones, where the direct effects of for instance heparin remain incompletely addressed. If complexes are formed between histones and heparin, these may cause the effects observed, but likewise the heparin itself may also be responsible for observed effects.

The design of the experiment was such that heparin was added ‘as a treatment’ after the addition of histones to whole blood. Heparin alone was also tested and fig 2 shows the effect of heparin only on inflammatory markers generated in whole blood in the absence of histones. The results do show heparin inhibited production of baseline inflammatory markers. The mixing of heparin-histones before incubation with blood will likely give a different response to adding heparin to blood pre- treated with histones. However, it is likely that in vivo histones would be circulating in blood before heparin would be given and hence, we have adjusted our experimental approach to reflect this likely in vivo setting.

For the fig 2, 3 and 4 the data has been corrected for the background response – this can be added?

The presentation of data corrected for background was considered the easiest to follow – following your suggestion the figures have been updated removing background correction and the responses for background and histone only have been added – statistical representation has been changed slightly to focus on this and values in the text have been updated to reflect this.

• In addition to this latter point, it has not been studied whether or not these complexes are formed, and only the formation of such a complex is inferred from the data, while it was not shown they actually exist, or that binding occurs and accompanies the effects measured.

This is an interesting point. Our observation is that heparin neutralises histones and we suggest, as others have, that this is due to direct, charge-based interaction between negatively charged heparin and positively charged histones. It has long been known that heparin binds to histones (references 22, 46, 47). We have shown in this study that the neutralisation of histones by heparin is dependent on the presence of negatively charged sulfate groups and that disruption of heparin’s charge distribution by exposure of amine groups removes its neutralising activity. This is consistent with our suggestion of direct interaction on the basis of charge. 

• It is unclear why a limited selection of inflammation biomarkers was chosen

We appreciate that a wide range of biomarkers could have been used. To ensure we obtain precise and reproducible results, we decided to carry out replicates and multiple dose response for each analyte and this meant we have to limit the number of biomarkers measured as we have a limited amount of sample from each treatment. We have chosen IL6 and IL8 as these are well known indicators of inflammations, tissue factor was chosen as this has been used to determine the effect of heparin on histones in vivo. With part of our interest on activation of the complement system C3a was chosen given its early generation in the complement cascade

• Anticoagulated blood is used, how did this affect the observations made, as this is non-physiological. No information on NET-formation or cell activation is provided.

Whilst we appreciate that this is a non-physiological assay, without an anticoagulant the blood will clot and prevent sample collection for analysis. The anticoagulant, hirudin was chosen to avoid the chelating of calcium which is important for various biological reactions, with it being suggested as a better anticoagulant for measurement of complement markers (Mollnes et al, Blood 2002) which was an area of interest in our study. NET formation and cell activation are complex scenarios that we should and will explore but we feel it is also important to study the relationship of individual components such as histones with heparin. 

Reviewer #2: In this review the authors us an in vitro approach with hirudin collected healthy whole blood to study the impact of heparins on the histone-induced inflammatory response.

A search of the literature did not reveal any similar in vitro experiments. There is some literature in this area in vivo. The current experiments are primarily descriptive.

Concerns

1. Why was the whole blood collected in hirudin? Would the same effects be seen if the blood was collected in citrate?

Hirudin was chosen as the anticoagulant as it does not chelate calcium, as with citrate and EDTA. Calcium is an important cation in various biological systems, including the complement system. Hirudin has also been suggested as a better anticoagulant for measurement of complement markers – Mollnes et al, Blood 2002.

2. The authors report low levels of cytokines and C3a in response to histone stimulation (ng/ml range). What is the detection range of the ELISA for these markers?

Thank you for this question – all samples were diluted to enable measurement of the analytes and the responses of the test samples were within the dose-response curves of the standards used. The detection ranges were: C3a 0.3 to 20 ng/ml, IL6 3.2 – 200 pg/ml, IL8 7.8 – 500 pg/ml and TF 0.4 to 100 u/ml

3. The introduction could be shortened and still convey the rationale for the study.

The introduction has been modified as suggested and we hope that it conveys the rationale for the work carried out.

4. Levels of markers should not be reported as negative. Perhaps best to report as percent reduction compared to control.

All figures have now been adjusted to include the background response for each analyte. We hope this makes it clearer to judge the ‘effectiveness’ of the different heparins.

5. The supplemental figure should be part of the main manuscript.

Thank you for this suggestion – this figure has been integrated into figure 2

6. The authors are missing a recent publication that supports their findings Zhu et al J. Trauma Acute Care Surgery 2019

Thank you for providing this reference – however, we are not sure whether this is the correct paper as it does not appear to relate to our topic. – Zhu et al ‘Shock index and pulse pressure as triggers for massive transfusion’ J Trauma Acute Care Surg V87, 1–S1 

7. Page 2 line 33 is missing a reference, Page 2 line 44 is this a complication of sepsis or pathogenesis (see McDonald et al Blood 2017)

Apologies, references have been added in. Our interpretation of the literature is that it may be part of the pathogenesis since high histones levels appear to correlate with high severity in sepsis. The histone levels may either be as a result of broad cellular damage from the infection or as a result of uncontrolled NETosis.

---

## [Decision Letter · Decision Letter 1]

11 May 2020

Heparin and non-anticoagulant heparin attenuate histone-induced inflammatory responses in whole blood

PONE-D-20-00191R1

Dear Dr. Hogwood,

We are pleased to inform you that your manuscript has been judged scientifically suitable for publication and will be formally accepted for publication once it complies with all outstanding technical requirements.

With kind regards,

Pablo Garcia de Frutos

Academic Editor

PLOS ONE

Additional Editor Comments (optional):

Reviewers' comments:

Reviewer's Responses to Questions

**Comments to the Author**

1. If the authors have adequately addressed your comments raised in a previous round of review and you feel that this manuscript is now acceptable for publication, you may indicate that here to bypass the “Comments to the Author” section, enter your conflict of interest statement in the “Confidential to Editor” section, and submit your "Accept" recommendation.

Reviewer #2: All comments have been addressed

2. Is the manuscript technically sound, and do the data support the conclusions?

Reviewer #2: Yes

3. Has the statistical analysis been performed appropriately and rigorously? 

Reviewer #2: Yes

4. Have the authors made all data underlying the findings in their manuscript fully available?

Reviewer #2: Yes

5. Is the manuscript presented in an intelligible fashion and written in standard English?

Reviewer #2: Yes

6. Review Comments to the Author

Reviewer #2: The authors have made all the suggested revisions including figures, rationale for experimental approach and statistical analysis.

7. PLOS authors have the option to publish the peer review history of their article (what does this mean?). If published, this will include your full peer review and any attached files.

Reviewer #2: Yes: Alison Fox-Robichaud

---

## [Editor Report · Acceptance letter]

15 May 2020

PONE-D-20-00191R1 

Heparin and non-anticoagulant heparin attenuate histone-induced inflammatory responses in whole blood 

Dear Dr. Hogwood:

I am pleased to inform you that your manuscript has been deemed suitable for publication in PLOS ONE. Congratulations! Your manuscript is now with our production department. 

With kind regards,

on behalf of

Dr. Pablo Garcia de Frutos 

Academic Editor

PLOS ONE